# Metabolic Background, Not Photosynthetic Physiology, Determines Drought and Drought Recovery Responses in C3 and C2 Moricandias

**DOI:** 10.3390/ijms24044094

**Published:** 2023-02-17

**Authors:** Carla Pinheiro, Giovanni Emiliani, Giovanni Marino, Ana S. Fortunato, Matthew Haworth, Anna De Carlo, Maria Manuela Chaves, Francesco Loreto, Mauro Centritto

**Affiliations:** 1UCIBIO Applied Molecular Biosciences Unit, Department of Life Sciences, NOVA School of Science and Technology, Universidade NOVA de Lisboa, 2829-516 Caparica, Portugal; 2Associate Laboratory i4HB Institute for Health and Bioeconomy, NOVA School of Science and Technology, Universidade NOVA de Lisboa, 2829-516 Caparica, Portugal; 3Institute for Sustainable Plant Protection, National Research Council of Italy (CNR-IPSP), Via Madonna del Piano 10 Sesto Fiorentino, 50019 Firenze, Italy; 4Instituto de Tecnologia Química e Biológica, Universidade Nova de Lisboa, Av. da República-EAN, 2780-157 Oeiras, Portugal; 5Institute of BioEconomy–National Research Council of Italy (CNR-IBE), Via Madonna del Piano 10 Sesto Fiorentino, 50019 Firenze, Italy; 6Instituto Superior de Agronomia, Universidade de Lisboa, Tapada da Ajuda, 1349-017 Lisboa, Portugal; 7Department of Biology, University of Naples Federico II, 80126 Naples, Italy

**Keywords:** C2-metabolic signature, C3-C4 intermediates, RNAseq, starch and sugars

## Abstract

Distinct photosynthetic physiologies are found within the *Moricandia* genus, both C3-type and C2-type representatives being known. As C2-physiology is an adaptation to drier environments, a study of physiology, biochemistry and transcriptomics was conducted to investigate whether plants with C2-physiology are more tolerant of low water availability and recover better from drought. Our data on *Moricandia moricandioides* (Mmo, C3), *M. arvensis* (Mav, C2) and *M. suffruticosa* (Msu, C2) show that C3 and C2-type *Moricandias* are metabolically distinct under all conditions tested (well-watered, severe drought, early drought recovery). Photosynthetic activity was found to be largely dependent upon the stomatal opening. The C2-type *M. arvensis* was able to secure 25–50% of photosynthesis under severe drought as compared to the C3-type *M. moricandioides*. Nevertheless, the C2-physiology does not seem to play a central role in *M. arvensis* drought responses and drought recovery. Instead, our biochemical data indicated metabolic differences in carbon and redox-related metabolism under the examined conditions. The cell wall dynamics and glucosinolate metabolism regulations were found to be major discriminators between *M. arvensis* and *M. moricandioides* at the transcription level.

## 1. Introduction

In the Mediterranean region, plants are challenged by frequent water shortages, often in association with high temperatures, impairing net photosynthesis (Pn) [1,2,3]. These conditions reduce the availability of CO_2_ at the ribulose-1,5-bisphosphate carboxylase-oxygenase (RuBisCO) active site due to stomatal closure, which, by reducing latent heat release, also makes leaves warmer. High temperatures also affect RuBisCO enzymatic properties, favoring oxygenation [4]. The combination of low water availability with high temperatures is considered an evolutionary driver for the CO_2_ concentrating mechanisms observed in CAM, C4- and C2-type species (aka C3-C4 intermediates; [5]). In C2-type species, the lack of a functional glycine decarboxylase in the mesophyll cells implies that glycine decarboxylation can only occur in the bundle sheath cells, resulting in a higher CO_2_ concentration near the RuBisCO active site. Thus, at a physiological level, C2-type species are characterized by lower CO_2_ compensation points than those of C3 species [6,7].

C2-type photosynthesis is considered to be a stable evolutionary state [8,9]. The transition from C3 to C4 photosynthesis has occurred in numerous plant lineages [10,11]. C2 species are typically found in warmer, more arid habitats than their C3 relatives [9,12]. Possible advantageous adaptations of C2 species may be more efficient water use and/or an increased ability to recover photosynthesis (Pn) when water becomes available. These can be useful traits in improving photosynthetic performance, since the genes required for C2 biochemistry are present in the majority of crop species, which have C3-type photosynthesis [9,13]. C2-type species are found in genera with or without C4 representatives, such as *Flaveria* and *Moricandia* [6]. The term “*Moricandia* syndrome” was originally used as a synonym for C2-type photosynthesis [14]. The lower CO_2_ compensation is accompanied by N-metabolite shuttles in some C2 species of the genus *Flaveria* [15,16], this feature being described as a C2-signature. Although it has been shown that C2- and C3-type *Moricandia* species are metabolically distinct, the presence of the C2-signature found in C2-*Flaveria* was not as clearly evident in C2-*Moricandia* (*M. arvensis*, *M. suffruticosa*) [17].

It has been proposed that the activation of C2-type photosynthesis will allow for sustained carbon gain under situations of high temperatures and limited water supply [18]. The balance between water saving via reduced transpiration and enhanced biochemical photosynthetic assimilation is particularly critical in semi-arid regions [19]. However, the nexus between photosynthesis, respiration and water efficiency (CO_2_ and water fluxes) under drought and recovery is largely unknown for C2-type photosynthesis. These plants do not minimize photorespiration to the same extent as C4 species; rather, CO_2_ released by photorespiration is recaptured by the ‘photorespiratory shuttle’ [20]. This should theoretically increase the ratio of photosynthesis relative to photorespiration and enhance the biochemical efficiency of CO_2_ assimilation [21]. However, there is a paucity of physiological data for robustly assessing the efficacy of the C2 photorespiratory shuttle in enhancing the carboxylation velocity of RuBisCO or the maximum rate of photosynthesis. Moreover, enhanced ratios of photosynthesis relative to photorespiration will also likely improve water use efficiency through enhanced CO_2_ assimilation relative to transpirative water loss. Despite observations of C2 species predominantly occurring in arid, drought-prone regions, to the best of our knowledge, there are no reports where the drought responses of closely related C3 and C2 intermediate species are robustly assessed. Given the capabilities of many plants to ‘switch on’ genes characteristic of C2 and C4 photosynthetic physiologies [22], it may be reasonable to presume that the expression of genes underpinning C2 photosynthesis may become more pronounced under drought conditions to enhance the photosynthetic carbon gain. This work aims to address these critical knowledge gaps in our characterization and understanding of C2 photosynthesis. It should be noted that the C2-shuttle will require metabolic adjustments of carbon and nitrogen metabolism, as well as reducing equivalents [6]. As *Moricandia* species have been shown to be metabolically distinct [17], such distinctions may highlight the specific regulatory functions of metabolism, namely, in carbon partitioning between soluble sugars and starch.

We evaluated the effects of increasing water deficit on *Moricandia* species with different photosynthetic physiologies. Our working hypothesis is that, in response to progressive soil water deficit, C2 *Moricandia* species will maintain photosynthetic activity at lower water availabilities and recover faster than their C3 counterparts when rewatered. Specifically, we assessed whether, under water deficit, the C2-metabolic signature is more pronounced than that under well-watered conditions, thus revealing phenotypic plasticity in C2-*Moricandia* species.

## 2. Results

### 2.1. Cross-Species Comparison under Control Conditions

#### 2.1.1. Physiology

Considering the physiological parameters analyzed (Figure 1), a large variability within each species and sampling point was observed; however, a gross separation between species may be identified.

An analysis of well-watered plants at day 0 (Figure 2, Appendix A) shows that the C2 species (Mav and Msu) significantly differed in all of the physiological parameters considered, except for the maximum rate of electron transport required for ribulose-1,5-bisphosphate (RuBP) regeneration (*J*max) and the carboxylation capacity of RuBisCO (*Vc*max).

The lowest photorespiration (PRP) was observed in Msu, and this was consistent with the observation of a lower CO_2_ compensation point in this species in comparison to that in Mmo and Mav. Using the curve-fitting method [23], we found that the mesophyll conductance to CO_2_ (Gm) was quite distinct for the C2 species: Mav displayed the highest Gm value, Msu showed the lowest value and Mmo exhibited an intermediate one (Appendix A).

#### 2.1.2. Biochemistry and Transcriptome Profiles

A distinct biochemical background under control conditions was revealed by multivariate analysis, with the species being separated along the first and second axis (Figure 3; Appendix A). Unlike the physiological analysis, a gross separation between C2 and C3 species occurs along the first axis.

A more detailed analysis at day 0 (Figure 4, Appendix A) highlighted major differences for the starch, sucrose and sucrose-to-hexose ratios. In addition, the redox equivalents (as estimated by the Folin–Ciocalteu method) also differed between the species, illustrating differences in the redox metabolism.

The comparative transcriptome analysis of *Moricandia* species can be achieved by making use of the minimal transcriptome of *Arabidopsis*, as previously described [17] (and the references therein). Although Mmo and Mav genome assembly is now available [24], it is not annotated. Therefore, it was considered preferable to use the highly annotated and curated *Arabidopsis* genome. Cross-species comparisons at the transcriptome level of all available well-watered controls showed a clear separation between the species; the C2-species (Mav and Msu) separate from each other along the second axis (Figure 5). The PCA analysis also revealed that, during the experimental time course, even under control conditions, a large effect of the sampling point and/or a large variability within the sampling point was observed. Additionally, Msu is a tetraploid [12,25], and it has been shown that genomic ploidy affects transcriptome assembly and quantification accuracy (e.g., [26]). Furthermore, we made use of the pipeline developed under the framework of the ‘3to4 consortium’ [17,27], in which the different ploidy level was not addressed (Nebion, personal communication). In this way, quantitative transcriptome analysis will focus on the cross-species comparisons between Mmo and Mav, which are both diploids.

The comparison of the Mmo and Mav transcriptome at day 0 (Figure 5B) shows that 11.8% of the transcriptome was differentially expressed. A contribution of the photosynthetic type and primary carbon metabolism to the discrimination between species was not evident. Indeed, a large majority of the genes related to CO_2_ concentrating mechanisms, the Calvin cycle and the C2 metabolic signature (as described by [17]) were not found to be differently expressed across the *Moricandia* species (only the expression of 4 out of 68 genes shows differences between Mav and Mmo; Appendix A). Considering the Biological Process Gene Ontologies, the main differences between the species were observed for the secondary metabolism, phenylpropanoid biosynthesis and cell wall organization. To perform a detailed analysis of these processes, we made use of selected KEGG pathways maps (Appendix A). The rationale was to combine these KEGG pathway maps with our biochemical findings regarding the cross-species differences found in redox equivalents and the starch and sucrose content (Figure 4). Specifically, we targeted the maps: ath00940 (Phenylpropanoid biosynthesis, 128 enzymes), ath01110 (Biosynthesis of secondary metabolites, 1280 enzymes), ath00480 (Glutathione metabolism, 39 enzymes), ath04146 (Peroxisome, including superoxide dismutases in the analysis) and ath00500, which includes all the enzymes involved in starch and sucrose metabolism (76 enzymes). For the selected pathways, the differently expressed genes (*p* < 0.01, log fold change > 1) between Mav and Mmo were considered (Appendix A).

Using this approach, we found 135 differently expressed genes at day 0 (61 genes were upregulated in Mmo; 74 genes upregulated in Mav). Most of the enzymes in the analysis are coded by multigenic families, and, unsurprisingly, we found that only some members of the gene family were differently expressed. However, with the exception of the largest gene families, a general consistency was found. This was the case for the genes of the starch and sucrose metabolism. The isoforms of sucrose synthase (*SUS6*, *SUS1*), trehalose synthase (*TPS8*), trehalose phosphatase (*TPPB*, *TTPE*), the enzymes of starch degradation (*PSH1*, *PSH2*, *BAM2*) and synthesis (*APL4*, *APS2*) were found to be more highly expressed in Mav than in Mmo. In contrast, in Mmo, there was a relatively higher expression of cell wall invertase *ATBFRUCT1* (a.k.a. *CWI1*) and *AMY1*, a secreted protein with alpha-amylase activity. Our analysis also showed differences in the expression of several β-glucosidases (*BGLU*) involved in sugar metabolism and cell wall remodeling [28]. In our dataset, differential expression was found for members of the subfamilies 5 and 10 (*BGLU27*, *BGLU31*, *BGLU32*, *BGLU45* upregulated in Mmo) and the subfamilies 1 and 6 (*BGLU11*, *BGLU33* upregulated in Mav). A higher expression of a number of genes related to redox metabolism was found in Mmo. At day 0, ten glutathione S-transferases (GST), six cell wall peroxidases (Px) and three superoxide dismutases (SOD) were upregulated in Mmo, while one GST, four Pxs and one ascorbate peroxidase (APX) were down-regulated. On the other hand, catalase was not found to be differently expressed between the two species (at day 0 or at any other sampling point).

### 2.2. Progressive Soil Water Deficit and Rewatering

#### 2.2.1. Physiological Evaluation

The water deficit treatment was imposed by withholding water at day 0 (T0). Photosynthesis (Pn) and stomatal conductance (G_s_) were recorded each day, as the soil dried, to reflect the ‘drought kinetic’ of the *Moricandia* species (Figure 6). The fraction of transpirable soil water (FTSW) technique serves as a relative scale based upon the start/end point G_s_ and the pot weight as the soil dries [29], allowing for a comparison of plants of differing sizes, species and pot sizes (e.g., [30,31]). It is noteworthy that the G_s_ values of Msu (the C2 species with strong C2-physiology on the basis of the CO_2_ compensation point and estimated photorespiration; Figure 2) were broadly lower than those of C2 Mav and C3 Mmo (Figure 6A). The drought kinetics in terms of the Pn and G_s_ response of C2 Mav and C3 Mmo were broadly identical; the internal CO_2_ concentration (Ci) decreased in Mmo and Mav, but not in Msu. Pn values were lower in C2 Msu once FTSW declined below 80%, although Ci did not decrease (Figure 6C). The decrease in Pn, G_s_ and Ci observed for Mm and Mav indicates the strong stomatal control of *P*_N_ (e.g., [32,33]). It is noteworthy that the foliar RWC of Msu when G_s_ fell to 10% of the values recorded at 100% FTSW were around 40%, while the RWC values of C3 Mmo and C2 Mav were ~60%.

The two C2 species showed a lower relative water content (RWC) at T2 (~10% of the starting G_s_) and maintained G_s_ at a lower leaf RWC than the C3 species. However, the three species showed quite severe drops in G_s_, as the plants went from full G_s_ to 50% to virtually zero stomatal conductance very rapidly.

#### 2.2.2. Cross-Species Comparison at the Physiological and Biochemical Levels

To perform a cross-species comparison, the data were normalized, i.e., the fold change to its respective control for each parameter was considered (Figure 7). Using this approach, and when considering the physiological and biochemical parameters, Mmo treatments separate along the first axis, revealing a strong response to drought and drought recovery. Mav treatments separated along the second axis, which explained a lower variation. Msu treatments separated along the first axis when considering the physiological data and along the second axis when considering the biochemical parameters. Considering all the physiological parameters available (Appendix A), Mmo was affected at T1 (~50–60% of the starting G_s_), while the C2 species showed alterations only at T2, with a few exceptions in Mav: G_s_, photorespiration was affected at T1; respiration in the light (Rd) and in the dark (Rn) was not affected by drought. The effect of stomatal closure was observed in the three species, i.e., Pn was highly responsive to G_s_. In this way, photosynthesis was largely controlled by stomata and increased the diffusive resistance to the uptake of CO_2_. It was also observed that photorespiration declined with Pn. Regarding recovery, Mav exhibited a faster recovery than Mmo and Msu in terms of G_s_ and Pn (Figure 6, Appendix A). The multivariate analysis of the biochemical data (Figure 7C; Appendix A) shows that, in both Mmo and Mav, T1 and RW grouped more closely than they did in T2, indicating a strong T2 effect that were not visible in rewatering. Our data show that, at T1, a higher number of parameters were affected in Mav (only sucrose, glucose and fructose were not affected) than in Mmo (only raffinose, a typical stress-responsive oligosaccharide, was affected [34]).

#### 2.2.3. Cross-Species Transcriptome Comparison at T2 and RW

As described in Section 2.1.2, quantitative transcriptome analysis focuses on the diploids Mmo and Mav. Establishing the comparison as Mmo vs. Mav, major differences between the two species were observed (Figure 8). At T2, 2456 genes show differential expression (*p* < 0.01, log fold change > 1; 1405 upregulated in Mmo; 1051 genes upregulated in Mav), while at RW, 2632 genes show differential expression (*p* < 0.01, log fold change > 1; 1226 upregulated in Mmo; 1406 genes upregulated in Mav). Several genes were found to be simultaneously upregulated at T2 and RW: 42 genes in Mav and 120 genes in Mmo.

Enrichment analysis (false discovery rate, FDR < 0.05) allowed us to further highlight major differences between species (Appendix A). Biological processes related to the photosynthetic physiologies of the *Moricandia* species were not over-represented. The analysis of specific genes and functional categories related to CO_2_ concentrating mechanisms, the Calvin cycle and the C2 metabolic signature for Mmo and Mav did not reveal major alterations (Appendix A), with the possible exception of the N shuttle, which is considered to act in C2-type photosynthesis [17]. Three of the four aspartate aminotransferase genes (upregulated in Mav) and one of the two dicarboxylate transporter genes (upregulated in Mmo) were found to be differentially expressed. The data also showed that a higher number of genes were differently affected at early recovery than at T2 (687 vs. 433), illustrating distinct responses in drought recovery. At rewatering (RW), differences were found in functional categories related to the carbohydrate metabolism (including sucrose), cell wall-related pathways, oxi-reduction process and secondary metabolism (Appendix A). As described in Section 2.1.2, we specifically targeted several KEGG pathways maps and compiled those genes that were differently expressed between Mav and Mmo in Appendix A (*p* < 0.01, log fold change > 1).

At T2 (severe drought), 151 genes of the selected pathways were differently expressed (94 upregulated in Mmo, 57 upregulated in Mav). We also found that 43 of these genes (~30%) were also differently expressed at T0 with the same trend (upregulated in Mav at T0 and T2, upregulated in Mmo at T0 and T2). Similar to T0, sucrose synthase (*SUS1*, *SUS6*) and trehalose-6-phosphate synthase (*TPS8*) were upregulated in Mav, while *CWI1* was upregulated in Mmo. Unlike T0, at T2, no differences in expression were found for enzymes related to starch biosynthesis or degradation. Similar to what was found at T0, several *BGLU* were differently expressed; more genes were upregulated in Mmo (*n* = 8) than in Mav (*n* = 3), namely, BGLU involved in lignification (*BGLU45-46*, subfamily 10 [28]). Redox-related genes were also found to be differently expressed. The GST expression profile was found to be highly distinctive at T0 (and at RW), and this was not observed at T2. On the other hand, the expression of superoxide dismutases (SOD) and peroxidases showed the largest differences between species at T2, as two SODs, five cell wall peroxidases and two ascorbate peroxidases (APX) were upregulated in Mmo (versus three cell wall peroxidases upregulated in Mav).

At RW, 153 genes of the selected pathways were differently expressed (84 upregulated in Mmo, 69 upregulated in Mav). The expression pattern of 56 genes (~37%) was equal between RW and T2. Sugar and starch-related enzyme expression was found to be similarly expressed in Mav and Mmo, with possible exceptions in the upregulation of aspartate aminotransferase (*APS1*, *APS3*) and trehalose-related metabolism (*TPS8*, *TPS11*) in Mav. Regarding the redox-related metabolism, the biggest differences were due to the upregulation of 12 glutathione S-transferases (GST), only in Mmo. Peroxidases, namely, cell wall peroxidases, did not exhibit a clear pattern (five cell wall peroxidases were upregulated in Mav; six cell wall peroxidases and one stromal peroxidase were upregulated in Mmo). A higher number of *BGLU* genes were also found to be upregulated in Mmo at RW.

Considering the selected pathways, our approach showed a clear distinction between Mav and Mmo, irrespective of the growing conditions, such distinctiveness relying on metabolic pathways other than photosynthesis, e.g., secondary metabolism.

## 3. Discussion

The three *Moricandia* species under study, grown at a near-optimal temperature for C3 photosynthesis, show distinct biochemical features that allow for a gross separation between C2 and C3 species. This probably reflects the origins of the genus *Moricandia*, as Mav belongs to the African clade and Mmo belongs to the Iberian clade [12]. As observed in previous studies [17], the C2 metabolic signature was not observed in our work. At the physiological level, it was not clear if the photorespiratory shuttle was operating. Under progressive drought, Pn was largely controlled by stomatal closure and increased diffusive resistance to the uptake of CO_2_ in Mmo and Mav. Our working hypothesis was that Mav C2-physiology would be active only when advantageous. Indeed, the C2-physiology is described as improving carbon assimilation and water use efficiency under drought conditions [18]. Such phenotypic plasticity has been observed in several facultative C2, C4 and CAM species, where the plants switch to these metabolic syndromes from C3 metabolism when coping with less favorable environmental conditions [35,36,37]. However, under drought and early rewatering, the C2-physiology did not become increasingly evident. We hypothesize that: (i) C2-physiology does not provide sufficient benefits to cope with drought when plants are grown at a near-optimal temperature for C3 photosynthesis; (ii) the drought conditions assayed were not favorable to the establishment/revelation of a C2 feature; or (iii) the functioning of the C2 pathway was affected by other factor(s). Regarding the last hypothesis, Winter et al. [38] found that the CO_2_ compensation point in Mav is affected by the nitrogen source. In addition, a metabolic blockage has already been observed in Bromlieads, where the shift from C3 metabolism to CAM was observed under drought [35,36] only when nitrate was absent in the growing media [39,40].

While the C2-signature was not evident, the biochemical and transcriptome characterization highlighted distinct metabolic backgrounds that were maintained at severe drought (T2) and early recovery (RW). Our approach discloses the potential role of several multigenic families in such distinctiveness, namely, those involved in sugar metabolism. Considering the role of sugars as growth substrates and signaling molecules, the coordinated function of several enzymes, namely, invertases and sucrose synthase, impacts the sucrose-to-hexose ratios and sucrose availability, and distribution needs to be balanced between growth and translocation in source leaves. Sucrose feeds several metabolic pathways such as respiration, starch, cellulose and callose synthesis. Several sucrose synthase mutants (*SUS*, *sus*) show altered starch accumulation patterns and higher sucrose synthase activity relating to a higher starch content [41]. While starch accumulation is considered to be related to sucrose synthase activity, the level of dependence is under discussion [41,42,43]. In the terms of sucrose-to-starch status, trehalose-6-P (T6P) is considered to be a major player. T6P is considered to act as a signaling and feedback regulator linking growth to carbon status [44,45]. The cross-species comparison between Mav and Mmo shows a lower sucrose content and a higher starch content in Mav. At a transcriptomic level, it co-occurs with the higher expression of *SUS1*, *SUS6* and trehalose-6-phosphate phosphatase (*TPPE* and *TTPB*). A higher starch content is compatible with higher *SUS* expression. On the other hand, a lower sucrose content is compatible with higher *SUS* expression (that hydrolyzes sucrose) and higher *TPP* expression, as the enzyme hydrolyzes T6P to trehalose, and strong and positive correlations between sucrose and T6P were found [44]. In addition, none of the *TPS* genes coding for catalytic active proteins (*TPS1* to *TPS4* [46]) were found to be differently expressed in our dataset, supporting differential levels of T6P due to TPP activity. All TPP genes from *Arabidopsis* (*TPPA*-*TPPJ*) are considered to be active [47,48], and in our dataset, the upregulation in Mav of *TPPE* (with a role in stomata activity via ABA signaling [49]) and *TTPB* (involved in the regulation of the number of leaf cells [48]) is observed. Nevertheless, we only found differential expression for *TPS8* (upregulated in *Mav*). *TPS8* belongs to a non-catalytic TPS class [46] and has possible regulatory functions associated with abiotic and biotic responses, although the role of these proteins remains elusive [50].

Taken together, the transcriptomics analysis in Mav showed a lower expression of *SUS1* and *SUS6* (two of the six sucrose synthase genes in *Arabidopsis* [41]) and a higher expression of *TPPB* and *TPPE* (two of the ten known trehalose-6-phosphate phosphatase genes in Arabidopsis [47]), which agrees with the lower sucrose content and the higher starch content. The distinct carbon status (sucrose-to-starch and sucrose-to-hexose ratios) will also impact the cell wall metabolism, as revealed by the differential expression of several β-glucosidases (*BGLU*) involved in the sugar metabolism, cell wall remodeling and stress responses [51]. BGLU can act intra- and extracellularly, being coded by a large gene family (48 *loci*). The several isoforms are grouped into subfamilies according to their phylogenetic distance and putative function [28,52]. In our dataset, 14 *BGLU* with a wide specificity for β-D-glucosides, hydrolyzing terminal non-reducing β-D-glucosyl residues and releasing β-D-glucose (EC 3.2.1.21), and two ABA-specific β-glucosidases, acting on ABA-glucose ester (EC 3.2.1.175 [53]), were found to be differently expressed between Mmo and Mav. It is assumed that members of different subfamilies exhibit distinct substrate specificities and, thus, subfamily functionalities do not overlap [28]. However, *BGLU33* (AT2G32860, a unique member of subfamily 6) is able to rescue *bglu18* mutants [54], and, thus, both BGLU33 and BGLU18 (AT1G52400, sub-family 3) can be involved in the degradation of the biologically inactive ABA-glucose ester (ABA-GE, [55]). ABA-GE is generally found in vacuoles or the apoplast and can be transported into the endoplasmatic reticulum from the apoplast [55,56]. BGLU18 has an endoplasmatic reticulum retention signal, its activity being associated with increased ABA levels under drought [55,57], and in our dataset, it was upregulated in Mmo at T2. On the other hand, *BGLU33*, acting in the vacuole, was found to be downregulated at T0 and RW in Mmo. These data indicate the distinct regulation of the ABA metabolism due to severe drought (T2) and early drought recovery (RW) in Mav and Mmo. Nevertheless, further functional characterization is necessary, as the discussion on substrate specificity and physiological functions within the enzyme family is still ongoing. Recent work highlights that, in addition to the typical myronisanes BGLU34–BGLU39 [28], BGLU18–BGLU33 are also able to hydrolyze coumarin glucosides such as scopolotin and have a dual role as *S*- and *O*-glucosidases [57]. Myrosinases turn biologically inactive glucosinolates into active cyanogens, such a breakdown being involved in abiotic stress responses [57]. Our approach highlights a greater role for glucosinolates in drought and early recovery responses in Mmo in comparison with Mav. In addition to the intracellular acting BGLU, extracellular BGLU45 and BGLU46 are also involved in cell wall lignification [28,58] and were found to be upregulated in Mmo. Taken together with the higher expression of several secreted peroxidases (Class III) in Mmo, it is indicative of a potential difference in the level of cross-linking between cell wall compounds [59], with Mmo potentiality showing reduced cell extensibility and growth. The higher expression of several cell wall peroxidases and glutathione *S*-tranferases (GST) in Mmo further supports our biochemical findings, showing a higher reducing potential in this species. The redox state is a primary modulator of developmental processes, and, in addition to its catalytic functions, GST can have a non-catalytic role by modulating the ratio of reduced-to-oxidized glutathione forms [60].

The GST expression patterns were found to be quite distinct at T0 and RW but not so evident at T2. GST is associated with multiple functions in the growth, development and stress responses; several GSTs are identified as binding proteins of auxin and cytokinin [60,61]. GST is coded by a large multigenic superfamily and can act in the cytosol, microsomes and mitochondria [60]. In our dataset, differences were found for the plant-specific GST classes Tau and Phi, with serine as the active site residue, also exhibiting peroxidase activity [61,62,63]. GSTs are grouped into classes that usually have different general substrate profiles [60], both Tau and Phi GST being considered to be involved in development and stress responses [61,62,64]. Tau GSTs are subdivided into several classes according to their putative functional roles [62], which include stress responses (subclass I) and GSTs with a low affinity to glutathione but an ability to bind RNA (subclass II). Phi GST roles are related to environmental responses, playing a role in the biosynthesis and transport of secondary metabolites [64]. These metabolites can act as antioxidants under stress but are also developmental regulators, as flavonoid modulators of auxin signaling [60]. Considering both the catalytic and non-catalytic roles of GST and GST’s connections with hormone homeostasis, the differential expression of several GST genes may contribute to the apparent metabolic distinctiveness within the studied *Moricandia* species.

## 4. Materials and Methods

### 4.1. Plant Material

Seeds were multiplied within the Institute of Plant Biochemistry, Heinrich Heine University, from the seed stocks provided by: Osnabrück Botanic Gardens (*Moricandia moricandioides* line 04-0393-10-00); Royal Botanic Gardens in Kew (*M. suffruticosa* line 0105433); IPK Gatersleben (*M. arvensis* line MOR1).

### 4.2. Plant Growth Conditions

Plants were grown and subjected to treatments in controlled conditions at CNR-IPSP laboratories in Sesto Fiorentino, Italy. *Moricandia moricandioides* (Mmo, C3), *M. arvensis* (Mav, C2) and *M. suffruticosa* (Msu, C2) seeds were placed in trays containing Amsterdam medium (a 9:1 mix of washed sand and compost) and were allowed to germinate. The trays were supplied with a commercially available nutrient solution (COMPO Concime Universale, NPK 7-5-7, B, Cu, Fe, Mn, Mo, Zn: COMPO Italia, Cesano Maderno, Italy) at free access rates and kept within a plant growth room with a day/night temperature of 28/24 °C. Metal-halide lights were used to maintain a photosynthetically active radiation (PAR) of 800 μmol m^−2^ s^−1^ for 12 h each day. After two weeks, the plants were potted into 1.5 dm^3^ square pots filled with Amsterdam medium and kept within the same growth room. The plants were watered to pot capacity every two days and were provided with a commercial liquid plant fertilizer (COMPO Concime Universale) each week to facilitate nutrient availability at free access rates. The plants were grown for four weeks prior to the imposition of water deficit.

### 4.3. Drought Treatment and Leaf Sampling

The fraction of transpirable soil water (FTSW) method [29] was used to gauge the kinetics of the soil drying of the *Moricandia* species with different photosynthetic physiologies. Thirty-five plants of equivalent sizes were selected for each species. On the evening prior to the imposition of water deficit, all of the plants were watered to pot capacity. The pots were then allowed to drain overnight, weighed the next morning and sealed within plastic bags to eliminate evaporation from the soil. The stomatal conductance (G_s_) of the plants was then recorded and assumed to represent 100% of the potential G_s_. The pots were weighed and the G_s_ was recorded every day. The water deficit treatment was imposed by withholding water from 15 of the pots, and in the remaining pots (controls), water was replaced each day, equivalent to the starting weight. The opening of the bags for approximately 20 min each day enabled the exchange of gases between the soil and the atmosphere and prevented the soil from becoming anoxic. When the G_s_ declined to 10% of the starting G_s_ value in the water deficit treatment or the pot weight remained constant for three days, it was considered that all water within the soil available for transpiration had been exhausted (i.e., 0% FTSW), and the FTSW was calculated as [29]:(1)$$\mathrm{FTSW}=\frac{{(\mathrm{PW}}_{\mathrm{daily}}{-\mathrm{PW}}_{\mathrm{final}})}{{(\mathrm{PW}}_{\mathrm{initial}}{-\mathrm{PW}}_{\mathrm{final}})}$$
where PW is the pot weight in grams. After 10% G_s_ had been reached in the water deficit treatment, the plants were re-watered to their starting weight. The relative water content was determined using the leaf in the eighth position from the apex of the plant using the method of [65]. The same five plants per treatment and species were used for gas exchange analysis throughout the experiment; these plants were then sampled at the end of the experiment at the re-watering stage (the leaves used for gas exchange analysis were not sampled for further analysis).

For biochemical and RNAseq analysis, leaves from *Moricandia* plants were collected from the control (T0) and water-stressed plants (T1, T2), and, after re-watering (RW), they were frozen in liquid nitrogen and kept at −80 °C. Four fully expanded leaves (starting from the fourth leaf from the apex of the plant) from five plants per species and treatment were collected (i.e., directly above the leaves collected for the relative water content determination). The samples were collected at midday (T0, T1, T2) or after re-watering, when the values of the soil water content in both groups were identical (RW).

### 4.4. Plant Gas Exchange

Leaf gas exchange analyses were performed using a LiCor Li6400XT plant photosynthesis system connected to a 6400-40 leaf chamber fluorimeter cuvette (LiCor, Lincoln, NE, USA). The biochemical efficiency of photosynthetic CO_2_ assimilation was assessed in well-watered plants though the analysis of the response of photosynthesis (Pn) to an increasing sub-stomatal concentration of CO_2_ (C_i_). The concentration of CO_2_ within the leaf cuvette was reduced to 50 μmol mol^−1^ for approximately 30 min to remove any stomatal diffusive limitations to Pn before the CO_2_ concentration was increased in stages every 3 to 4 min, when Pn had stabilized (CO_2_ concentration steps: 50, 100, 200, 300, 400, 600, 800, 1000, 1200, 1400, 1600, 1800, 2000 μmol mol^−1^). The leaf temperature remained at 25 °C and the relative humidity remained at 60% throughout the Pn/C_i_ response curve [66]. The maximum carboxylation rate of RubisCO (*V*cmax) and the maximum rate of electron transport for the regeneration of RuBP (*J*max) were calculated using the curve fitting approach of Ethier and Livingston [23]. Due to the rapid stomatal closure of the *Moricandia* species [67], it was not possible to assess the response of Pn to C_i_ in plants subject to water deficit [68]. The CO_2_ compensation point was considered to be the point where the photosynthetic gain was zero. Point measurements of photosynthesis (Pn), stomatal conductance (G_s_), the sub-stomatal concentration of CO_2_ (C_i_) and the actual quantum efficiency of photosystem II (ΦPSII) [69] were undertaken throughout the FTSW response on the second-uppermost fully expanded leaf (position five from the leaf apex) of five plants per treatment and species. The following environmental conditions were set in the cuvette: 1000 μmol m^−2^ s^−1^ PAR (10% blue and 90% red light), 400 ppm (CO_2_), a leaf temperature of 30 °C and a relative humidity (RH) of 60%. To reduce diffusive leaks through the chamber gasket, a supplementary gasket was added, and the IRGA exhaust air was fed into the space between the chamber and the supplementary external gasket [70]. To determine ΦPSII, the multi-phase fluorescence setting was used, with an initial saturating pulse of 8000 μmol m^−2^ s^−1^ [71].

In C3 species, photorespiration can be calculated following [72]. However, for the two C2 *Moricandia* species, this approach is not appropriate, as a greater proportion of the previously released CO_2_ is recycled, preventing the accurate determination of the chloroplastic concentration of CO_2_ (C_c_) and mesophyll conductance. It should be noted that the Laisk method [73] for determining the photo-compensation point (*Γ**) where the CO_2_ consumed by Pn is released by photorespiration (P_PR_) is not applicable to C2 species. To determine P_PR_ and the possible role of the photorespiratory shuttle in the drought response of the three *Moricandia* species, Pn was therefore measured under ambient and 1.5% O_2_ (considered to represent non-photorespiratory conditions), which was shown to correlate to P_PR_, as determined using the Sharkey formula [21,74]. The use of O_2_ concentration effects on Pn to approximate P_PR_ is subject to error due to the recycling of respired CO_2_ [75] and the reversed oxygen sensitivity associated with limitations to Pn [76]. The O_2_ concentration of the air-stream entering the Li6400 was controlled using mass flow controllers (Brooks Instruments, Hatfield PA, USA) connected to compressed cylinders of O_2_ and N_2_. The mesophyll conductance to CO_2_ (Gm) was estimated by the curve-fitting method, a common approach for C3 plants [23].

Respiration in the light was determined using the method of [75], where the CO_2_ entering the cuvette is switched between CO_2_ composed of the heavier ^13^C and the lighter ^12^C isotope utilizing mass flow controllers. The efflux of ^12^CO_2_ is then determined using a ^13^CO_2_-insensitive infra-red gas analyzer (Gas-hound; Li-Cor) connected to the match valve tube of the Li6400. Respiration in the dark (Rn) was determined by switching the lights off within the leaf cuvette (i.e., 0 µmol m^−2^ s^−1^ PAR) and then recording the efflux of CO_2_ from the leaf after a period of 10 to 15 min.

### 4.5. Biochemical Analysis

Before the biochemical analyses, frozen samples were ground to a fine powder in liquid nitrogen and lyophilized in a SpeedVac^®^ Plus (Savant, Thermo Fisher Scientific, Waltham, MA, USA).

### 4.6. Estimation of the Reducing Capacity by the Folin–Ciocalteau Method

While the Folin–Ciocalteau method is typically described as a method for determining total phenolics, this method overestimates the soluble phenolics content since it also detects a range of reducing compounds, such as ascorbate and reducing sugars. Therefore, the method is a better proxy for the reducing capacity of the sample, to which phenolics contribute [77]. The extraction of the *Moricandia* leaves was performed according to [78], with slight modifications. Briefly, 100 µL of methanol was added to 10 mg of ground leaf material, left for 2 h on ice, with agitation, and centrifuged at 4 °C for 30 min at 16,100× *g*. The supernatant was removed, and two additional extractions were made. The three supernatants were combined, and the reducing capacity was quantified by the Folin–Ciocalteau method [79], with modifications. Briefly, to each microplate well, 235 µL of water, 5 µL of the sample, 15 µL of Folin–Ciocalteau’s reagent and 45 µL of saturated Na_2_CO_3_ aqueous solution were added. The microplate was then incubated at 40 °C for 30 min, and the absorbance at 765 nm was measured. Gallic acid was used as the standard, and the results were expressed in mg of gallic acid equivalents per gram of dry weight.

### 4.7. Extraction and Quantification of Water-Soluble Carbohydrates and Starch

Water-soluble carbohydrates were extracted following the addition of chloroform/methanol, as described [80]. Briefly, 10–15 mg of dried plant material was extracted with 250 µL of ice-cold chloroform: methanol (3:7, *v*/*v*), vortex-mixed and incubated for 2 h at −20 °C. After incubation, the samples were extracted three times with 200 µL of ice-cold water. After centrifugation, the upper phases were pooled and evaporated until dry (CentriVap Concentrator, Labconco, Kansas City, MO, USA). The samples were then reconstituted in 50 µL of water. Glucose, fructose and sucrose were quantified using a Sucrose/D-Glucose/D-Fructose kit (Cat N° 10716260035, R-BIOPHARM, Roche), while galactose and raffinose were quantified with the Raffinose kit (Cat N° 10428167035, R-BIOPHARM, Roche). We made use of the Hatterscheid and Willenbrink modification, which involved the oxidation of NADPH and the reduction of INT (p-iodonitrotetrazolium violet), catalyzed by diaphorase and measured at 490 nm [81].

The pellet resulting from the chloroform: methanol extraction was washed twice with 200 µL of water after being evaporated until dry, as described above. To the dry pellet, 500 µL of water and 3.13 µL of α-amylase (Sigma A4582, Rahway, NJ, USA) were then added to the pellet, which was boiled for 3 min and incubated at 121 °C for 50 min. After cooling, the samples were incubated for 2 h (pH 4.8 in 0.15 M sodium acetate buffer) at 60 °C with 2.2 µL of amyloglucosidase (1.7 mg/mL) (Sigma A7420). Starch was quantified in the gelified supernatant with the Starch kit (Cat N° 10207748035, R-BIOPHARM, Roche) and the Hatterscheid and Willenbrink modification.

### 4.8. RNA Isolation, RNA-Seq and Bioinformatic Analysis

Frozen samples were ground to a fine powder in liquid nitrogen, and the total RNA was isolated using the RNeasy Plant Mini Kit (Qiagen, Hilden, Germany), according to the manufacturer’s protocol, and digested with DNAse I using an on-column DNAse, as per the manufacturer’s instructions. Three to five replicates were considered per species and per sampling point. The intactness of the extracted RNA was verified by 1% agarose-TBE gel electrophoresis of the total RNA concentration, and the purity was verified through NanoDrop ND-1000 spectrophotometer (Thermo Fisher Scientific, USA) measurements.

After quality control on a Bioanalyzer 2100 (Agilent, Santa Clara, CA, USA), RNASeq was performed on an Illumina HiSeq2000 platform at Heinrich Heine University (Biologisch-Medizinisches Forschungszentrum, Düsseldorf, Germany). Sequence assembly and expression statistics were performed by Nebion AG (Zurich, Switzerland); all the *Moricandia* RNAseq reads were aligned to a minimal set of coding sequences of the TAIR 10 release of the *Arabidopsis* genome (http://www.arabidopsis.org/, accessed on 20 December 2022) using BLAT [82] in protein space. The mapping used BLAT with the options ‘−t = dnax −q = dna’. TPM (transcript per million) values and counts were computed from the best-scoring mappings only: if reads mapped to multiple features, only the best mappings were kept. The estimated counts data were imported in the DESeq2 R package [83] to perform further data quality assessment, exploratory analyses and transformation and to compute differential expression analyses. As some samples did not pass the quality control and were not submitted to RNAseq, the number of replicates for Mav at the T1 treatment was insufficient for further statistical analysis. The data discussed in this publication have been deposited in NCBI’s Gene Expression Omnibus [83] and are accessible through the GEO Series accession number GSE210653 (https://www.ncbi.nlm.nih.gov/geo/query/acc.cgi?acc=GSE210653, accessed on 13 February 2023).

Functional annotation, i.e., Gene Ontology assignment, was performed using the DAVID annotation tool (https://david.ncifcrf.gov/summary.jsp, accessed on 16 September 2021, [84]). Functional enrichment analysis was performed using: Profiler (https://biit.cs.ut.ee/gprofiler/gost, accessed on 27 September 2021, [85]) with a 0.05 FDR cut-off, providing a custom reference. Lastly, KEGG mapping was performed using the ‘KEGG mapper’ tool (https://www.genome.jp/kegg/mapper/, accessed on 4 July 2022, [86]).

### 4.9. Statistical Analysis

The statistical analysis was carried out on the R platform, v3.6.1. One-way ANOVA was applied to the data using the R function aov. Whenever significant differences between extracts were found, the honestly significant difference (HSD; *p* < 0.05) was calculated using the tukeyTest function from the PMCMRplus package. For principal component analysis and Pearson’s correlation, the ade4TkGUI package and the cor function were used [87,88].

## 5. Conclusions

The C3 and C2 *Moricandia* species were grown at a near-optimal temperature for C3 photosynthesis. The findings of this study indicate distinct adaptive capacities to cope with drought within the *Moricandia* genus, although not a prominent role for C2-physiology. At a physiological level, *M. moricandioides* responded more rapidly to drought, while, at the biochemical level, *M. arvensis* showed an earlier adjustment to decreasing soil water availability. A potential role for glucosinolates in drought and drought recovery was evidenced, and, at the transcriptional level, a comparatively higher contribution was found for *M. moricandioides* than for *M. arvensis*. Differences between the species in sugar, redox and cell wall remodeling are constants irrespective of growth conditions; transcriptomic and biochemical data pointed out reduced growth in *M. moricandioides*. Our approach demonstrates the relevance of the genotype and not the pathways related to C2-physiology that are comparatively indistinct. Overall, the kinetics of the drought responses of *Moricandia* species showed that the C2-physiology does not confer increased drought tolerance under the tested conditions. Conversely, water deficit caused metabolic differences in carbon and redox-related metabolism between the C3 and C2 species, although the mechanisms that produce these different metabolic responses remain unknown.

## Figures and Tables

**Figure 1 ijms-24-04094-f001:**
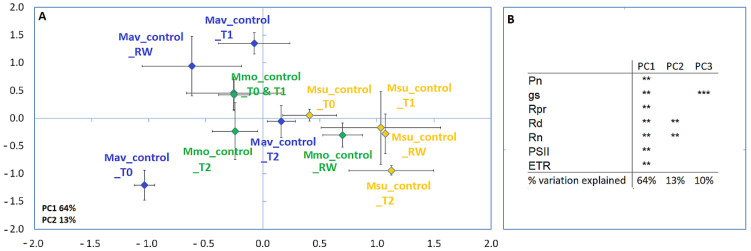
Principal component analysis (**A**) and Pearson correlations (**B**) for *M. moricandioides* (Mmo, green), *M. arvensis* (Mav, blue) and *M. suffruticosa* (Msu, orange) under control conditions and using the physiological data (** *p* < 0.01, *** *p* < 0.001). Input data are available as Appendix A. T0, T1, T2 and RW correspond to the several sampling points along the assay: T0—before the stress treatment; T1—controls for the plants under 50% FTSW (fraction of transpirable soil water); T2—controls for the plants under 10% FTSW; RW—controls for the plants submitted to drought (10% FTSW) and rewatered for 1 day.

**Figure 2 ijms-24-04094-f002:**
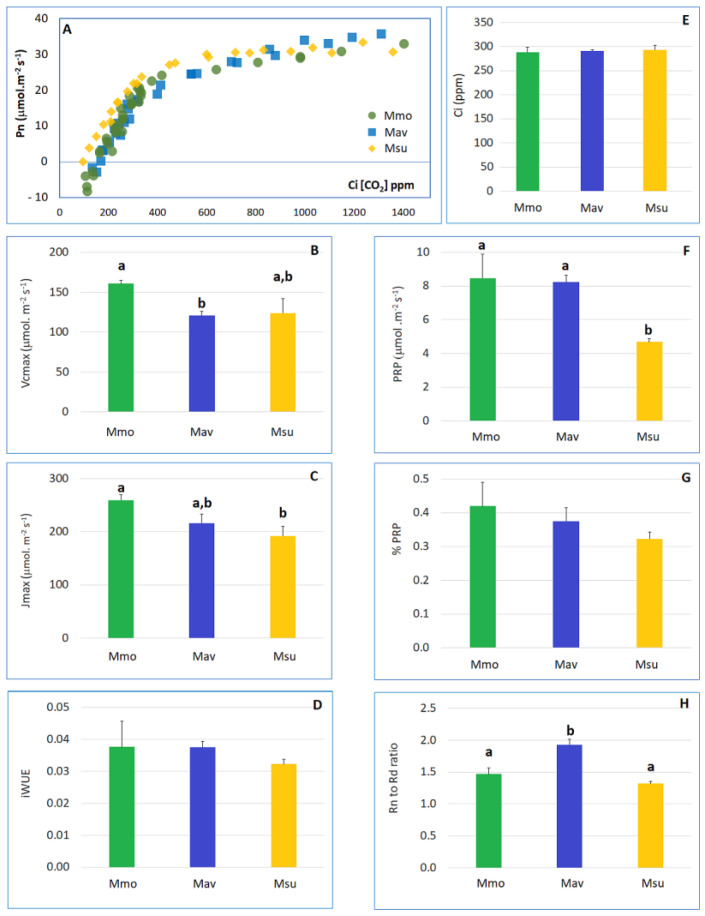
Detailed physiological analysis for the three species under well-watered conditions at day 0 (T0). (**A**) Leaf photosynthetic CO_2_-response (photosynthesis vs. internal CO_2_ concentration curves). (**B**) Maximum carboxylation rate of RuBP (Vcmax). (**C**) Maximum rate of electron transport required for RuBP regeneration (Jmax). (**D**) Intrinsic water use efficiency (iWUE), calculated as the Pn (μmol CO_2_ m^−2^ s^−1^) to stomatal conductance (G_s_, mmol CO_2_ m^−2^ s^−1^) ratio. (**E**) Internal CO_2_ concentration (Ci, ppm). (**F**) Photorespiration (P_RP_). (**G**) Ratio between photorespiration (P_RP_) and photosynthesis (Pn). (**H**) Ratio between night respiration (Rn) and diurnal respiration (Rd). Following One-Way ANOVA, and when significant differences were found, comparisons between species were assessed with Tukey’s HSD test. Different letters (a, b) indicate significant differences at *p* < 0.05. For Ci, iWUE and % PRP, no significant differences between species were found.

**Figure 3 ijms-24-04094-f003:**
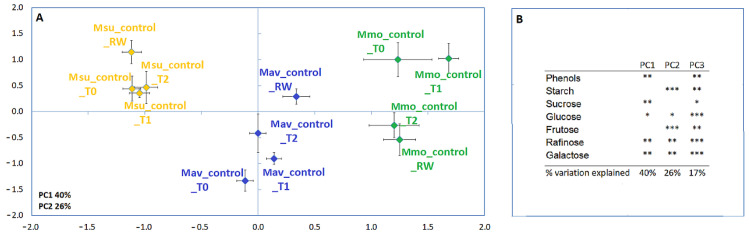
Principal component analysis (**A**) and Pearson correlations (**B**) for *M. moricandioides* (Mmo, green), *M. arvensis* (Mav, blue) and *M. suffruticosa* (Msu, orange) under control conditions and using the biochemical data (* *p* < 0.05, ** *p* < 0.01, *** *p* < 0.001). Input data are available in Appendix A. T0, T1, T2 and RW correspond to the several sampling points along the assay: T0—before the stress treatment; T1—controls for the plants under 50% FTSW (fraction of transpirable soil water); T2—controls for the plants under 10% FTSW; RW—controls for the plants submitted to drought (10% FTSW) and rewatered for 1 day.

**Figure 4 ijms-24-04094-f004:**
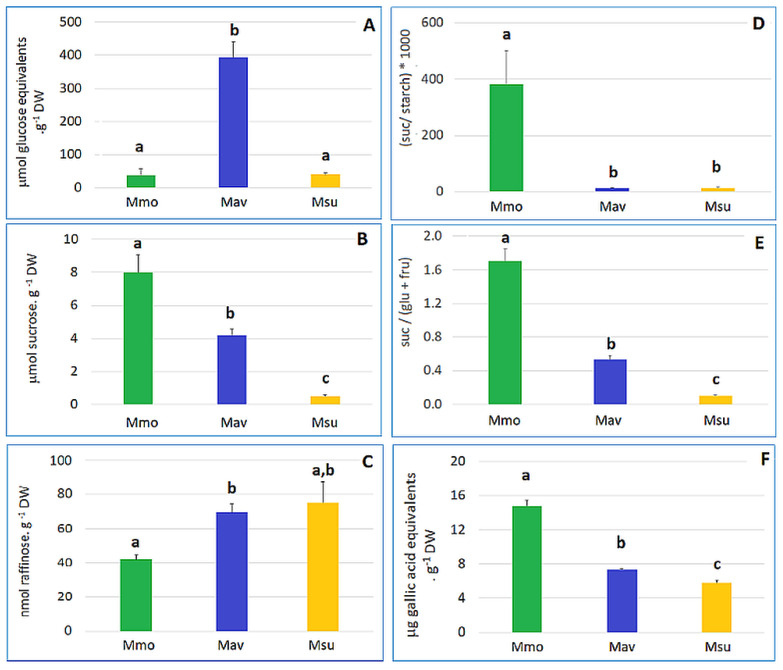
Starch, sugar and phenolics contents for the three species show a distinct biochemical status at day 0 (T0). (**A**) Starch. (**B**) Sucrose. (**C**) Raffinose. (**D**) Sucrose-to-starch ratio. (**E**) Sucrose-to-(glucose + fructose) ratio. (**F**) Phenolics. Cross-species comparison was performed with One-Way ANOVA, and when significant differences were found, a comparison between species was assessed with Tukey’s HSD test. Different letters (a, b, c) indicate significant differences at *p* < 0.05.

**Figure 5 ijms-24-04094-f005:**
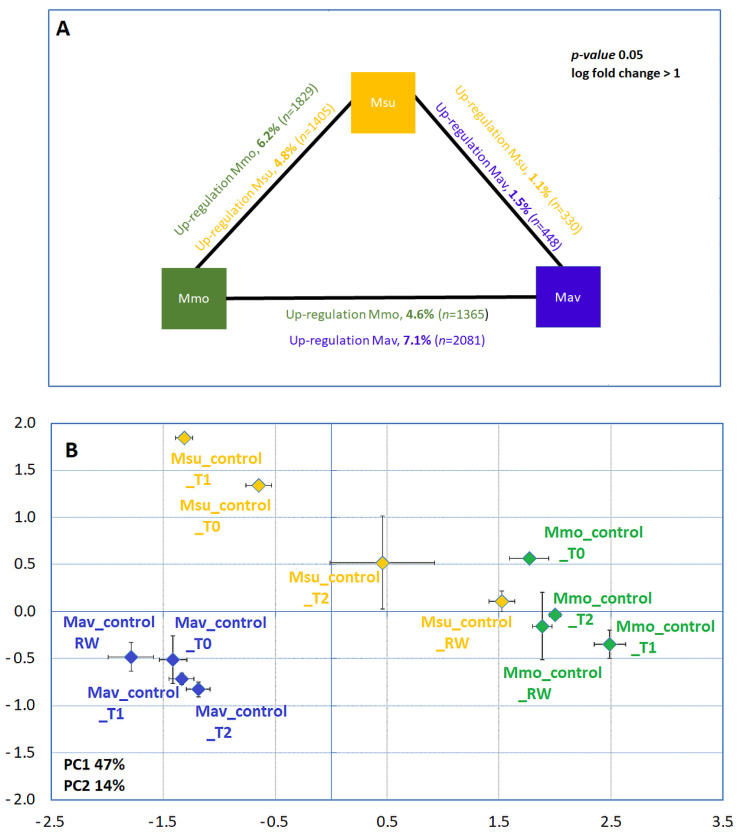
Cross-species gene expression comparison. (**A**) Differently expressed genes under control conditions at T0. (**B**) Principal component analysis under control conditions along the assay using the RNAseq data. (**B**). *M. moricandioides* (Mmo, green), *M. arvensis* (Mav, blue) and *M. suffruticosa* (Msu, orange) gene differential expression (*p* < 0.05, log fold change > 1). Within each species, and when comparing the available controls with T0, differently expressed genes were below 0.4% in Mmo and Mav and between 0.5 and 2.7% in Msu. RNAseq data are available at GEO (GSE210653). T0, T1, T2 and RW correspond to the several sampling points along the assay: T0—before the stress treatment; T1—controls for the plants under 50% FTSW (fraction of transpirable soil water); T2—controls for the plants under 10% FTSW; RW—controls for the plants submitted to drought (10% FTSW) and rewatered for 1 day.

**Figure 6 ijms-24-04094-f006:**
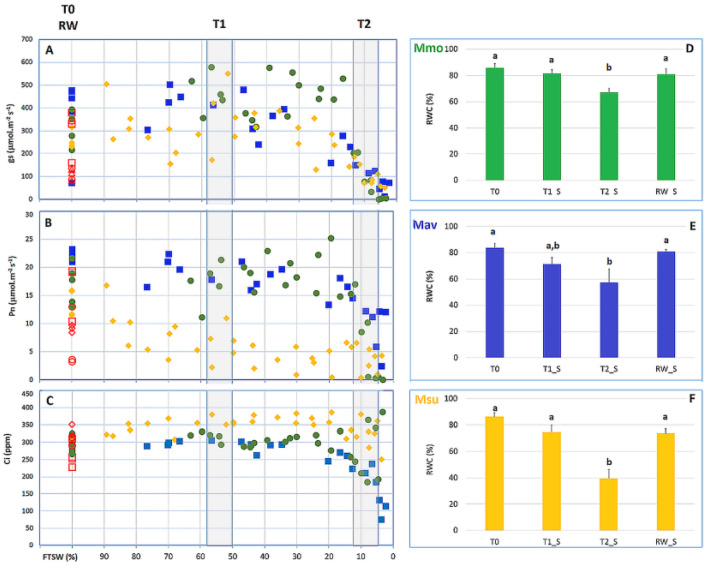
*M. moricandioides* (Mmo, green circles), *M. arvensis* (Mav, blue squares) and *M. suffruticosa* (Msu, orange diamonds) responses to progressive soil drying at the leaf level. (**A**) Stomatal conductance (G_s_); (**B**) Photosynthesis (Pn); (**C**) Internal CO_2_ concentration (Ci); (**D**) Mmo relative water content (RWC); (**E**) Mav RWC; (**F**) Msu RWC. For RWC, a comparison within each species was performed using a One-Way ANOVA, and when significant differences were found, comparisons within species were assessed with Tukey’s HSD test. Different letters (a, b) indicate significant differences at *p* < 0.05). Rewatering (100% FTSW) is represented by red empty symbols. Shaded areas in (**A**–**C**) represent the sampling points T1 and T2.

**Figure 7 ijms-24-04094-f007:**
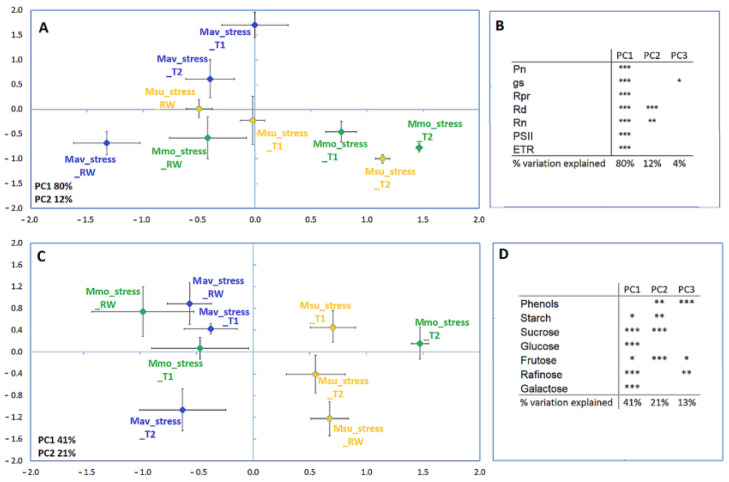
Principal component analysis (**A**,**C**) and Pearson correlations (**B**,**D**) for *M. moricandioides* (Mmo, green), *M. arvensis* (Mav, blue) and *M. suffruticosa* (Msu, orange) along the assay and using the physiological (**A**,**B**) biochemical data (**C**,**D**) (* *p* < 0.05, ** *p* < 0.01, *** *p* < 0.001). Input data are available in Appendix A.

**Figure 8 ijms-24-04094-f008:**
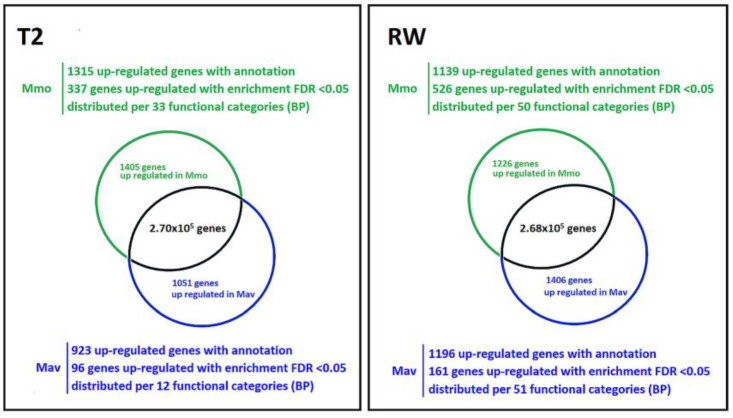
Cross-species transcriptomic analysis at T2 and RW (*p* < 0.01, log fold change > 1). Input data are available in Appendix A and GEO (GSE210653).

## Data Availability

The RNASeq data discussed in this publication are accessible through the GEO Series accession number GSE210653 (https://www.ncbi.nlm.nih.gov/geo/query/acc.cgi?acc=GSE210653, accessed on 13 February 2023).

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
