# Peer review of "Metabolic Background, Not Photosynthetic Physiology, Determines Drought and Drought Recovery Responses in C3 and C2 Moricandias"

_ijms, 2023, doi:10.3390/ijms24044094_

Round 1
Reviewer 1 Report
Dear Authors
Well done, I enjoyed reading your submission. I just wonder if you tested different Moricandia species with your desired soil-water deficiency prior to the main treatments as a pretest for checking their phenotypic characteristics?
Regards

Author Response
Well done, I enjoyed reading your submission.
Thanks! We try really hard to make the ms clear and focused.
Q. I just wonder if you tested different Moricandia species with your desired soil-water deficiency prior to the main treatments as a pre-test for checking their phenotypic characteristics?
A. The species to be tested were received under the framework of the 3to4 project (as acknowledged in the ms). After an initial screening by the Dusseldorf team (https://doi.org/10.1093/jxb/erw391), the 3to4 consortium selected these 3 species accessions to further evaluate its drought responses.
Our team performed many tests in order to find the right way to get them germinated and to properly monitor drought responses.
Reviewer 2 Report
Improve the title, to something like: "Evaluate the metabolic and physiological effect of water deficit in Moricandia species"
Improve the keywords, for example: C3-C4 intermediates; Drought; Photosynthetic activity; metabolic behavior
The sections of the article should be reordered: Introduction, Materials and Methods, Results, Discussion, Conclusions and References
In materials and Methods: Where and under what conditions was the study done?; indicate the amount of nutrients used; Based on what measurements was the amount of irrigation water calculated?; indicate the amounts of irrigation water applied to each treatment;
In Results: All symbols and abbreviations must be described within the text
In Conclusions: Further conclusions could be reached based on the results obtained.
Author Response
Q. Improve the title, to something like: "Evaluate the metabolic and physiological effect of water deficit in Moricandia species"
A. Following the reviewer's suggestion we discuss the title and think the current title adequately describes the findings of the work.
Q. Improve the keywords, for example: C3-C4 intermediates; Drought; Photosynthetic activity; metabolic behaviour
A. We adjusted keywords to “C2-metabolic signature”; “C3-C4 intermediates”, “RNAseq”, “Starch and sugars”. “Drought“ and “Photosynthetic activity” are part of the title and so not added as keywords.
Q. The sections of the article should be reordered: Introduction, Materials and Methods, Results, Discussion, Conclusions and References
A. Sections are ordered according to the journal guidelines.
In materials and Methods:
Q. Where and under what conditions was the study done?
A. Added in section 4.2. “Plants were grown and subjected to treatments in controlled conditions at CNR-IPSP laboratories in Sesto Fiorentino, Italy.”
Q. Indicate the amount of nutrients used;
A. Added in section 4.2. “… at free access rates”.
Q. Based on what measurements was the amount of irrigation water calculated?
A. As described in the ms (section 4.3), the fraction of transpirable soil water (FTSW) method (Australian Journal of Plant Physiology 13(3) 329 - 341). Pots were weighted every day and for control pots, water was added to the equivalent of the starting weight. For treatment pots, no water was added until rewatering.
Q. Indicate the amounts of irrigation water applied to each treatment;
A. The amount of water lost daily by each control pot ranged between about 30 and 40 ml, with no significant difference between the species. If necessary, we can add to the ms "for control pots, water was added to the equivalent of the starting weight (on average 30-40 ml per day)".
Q. In Results: All symbols and abbreviations must be described within the text.
A. We went through the ms and now provide a description of symbols and abbreviations at the time of its first appearance (RuBisCO, Pn, FTSW, Jmax, VCmax, Ci, Gm, T0, Gs, PRP, T1, T2, RWC, Rd, Rn, FDR, SOD, APX, GST, ER, PSII, TPM). Most of them were fully described in the Material and Methods section that, according to the journal guidelines, is presented after Results and Discussion. At the time of submission, we didn’t realise it and thanked the reviewer for the opportunity to correct it.
Q. In Conclusions: Further conclusions could be reached based on the results obtained.
A. Some details were added to the conclusions and we hope they respond adequately to the reviewer's comment.
Reviewer 3 Report
The article is very good and novelty, just carfully check the English Spelling
Author Response
Thanks! We try really hard to make the ms clear and focused.
The text was carefully revised for all authors, including Matthew Haworth, which is a native English speaker.
Reviewer 4 Report
The manuscript entitled “Metabolic background, not photosynthetic physiology, determines drought and drought recovery responses in C3 and C2 Moricandias´s” provides insight into the biochemistry of C2 plants. The manuscript is well organized, and all described methods are applied according to the high scientific standards. Therefore, in my opinion, the manuscript can be accepted in its present form.
Author Response
Thanks! We try really hard to make the ms clear and focused.